# Reply to Horacek, M.; Cannavan, A. Comment on “Sinkovič et al. Isotope Fingerprints of Common and Tartary Buckwheat Grains and Milling Fractions: A Preliminary Study. *Foods* 2022, *11*, 1414”

**DOI:** 10.3390/foods11172628

**Published:** 2022-08-30

**Authors:** Lovro Sinkovič, Nives Ogrinc, Doris Potočnik, Vladimir Meglič

**Affiliations:** 1Crop Science Department, Agricultural Institute of Slovenia, Hacquetocva Ulica 17, SI-1000 Ljubljana, Slovenia; 2Department of Environmental Sciences, Jožef Stefan Institute, Jamova Cesta 39, SI-1000 Ljubljana, Slovenia; 3Jožef Stefan International Postgraduate School, Jamova Cesta 39, SI-1000 Ljubljana, Slovenia

We thank Dr. Horacek and Dr. Cannavan for their interest [1] in our paper and feedback [2]. Regarding the naming of the different milling fractions of buckwheat grains, several terms are commonly used for the same fraction, and no precise definitions exist. In our paper, we used the terms “hulls” and “light flour”, as used in Steadman et al. [3,4]. Below are some further explanations of buckwheat stone milling used in our study.

Grain milling is probably the oldest manufacturing process in the world, with specific dry and wet milling techniques used in the food industry [5]. Milling begins with the physical crushing of the buckwheat grain, followed by the separation of the endosperm from the bran and pericarp [3]. Four main methods are used for milling buckwheat grains, and the chosen method significantly affects the properties of the obtained fractions: stone, roller, ultrafine, and wet milling [6]. Stone milling has distinct advantages, such as ease of use and simplicity of the system, higher content of macro/microelements and polyphenols in the flour compared to the other milling methods, and popularity among consumers [5].

Buckwheat seed milling fractions are obtained by the milling process in which the hulls are separated from the buckwheat grains. The results of further milling are usually light flour and grits/semolina, sometimes also bran fractions. Traditional milling is performed by using stone mills, which usually have a lower milling capacity and are suitable for smaller producers. Mill fractions contain varying proportions of central endosperm, embryo, and maternal tissues, which may vary in composition. Light flours contain mainly central endosperm, grits/semolina are hard chunks of endosperm, while bran contains seed coat and embryo tissue [3,4]. Since fatty acids in the grains are not distributed uniformly, the type of tissue contained in each fraction directly affects its composition. Light flour contains mainly the central endosperm and about 1% lipids [3]. The lipid content also varies in flours obtained by different milling methods [6]. The flour yield for common and Tartary buckwheat when milling by a traditional stone mill is similar [7].

*How were the milling fractions in our study prepared?* In Slovenia, the hulls are not removed before milling when a traditional stone mill is used. The milling process was performed using a traditional cereal stone mill, with a capacity of approx. 4 kg/h for buckwheat grain. Clean, undamaged whole grain underwent successive milling and sieving using different metal and polyester sieves, to obtain the light flour, semolina, and hulls. Semolina mostly represented the particles of the crushed embryos without hulls that remained in the millstone. The proportions of the fractions obtained after milling were ~60% light flour, ~30% hulls, ~6% semolina, and ~4% milling losses for the common buckwheat, while ~50% light flour, ~40% hulls, ~5% semolina, and ~5% milling losses for the Tartary buckwheat.

The *δ*^13^C values reported in the paper, commented on by Dr. Horacek and Dr. Cannavan, refer only to the lower *δ*^13^C values observed in buckwheat and not to isotopic compositions in general. Since the isotopic signatures of C3 and C4 plants are well-known and given the paper’s target audience, we assumed there was no need to provide a lengthy discussion in the paper but rather comment on the distribution of C isotopes concerning other studies on wheat samples. Perhaps, the sentence would better be rewritten as *“δ^13^C values in plants can also be affected by water availability, stomatal conductance and water use efficiency [8].”*

Regarding the interpretation of the conventional and organic samples by the farming system used, we agree with Dr’s Horacek and Cannavan that different locations with different environments and soil conditions could affect *δ*^15^N and *δ*^13^C values. In our view, the highest *δ*^15^N value of 7.8 ‰ found in the Dolenjska region could not be solely explained by the local environment and soil conditions. Our previous investigation performed on soil overlying carbonate bedrock from the Dolenjska region had *δ*^15^N values < 5‰ [9]. We agree that our results are indicative, but we believe these are of value for discussion in future similar studies.

We also would like to thank the authors for their thought on the *δ*^34^S values, which may open some further discussion. For instance, very low *δ*^34^S values (below ca. −20‰) were found in grain from Tuscany in the study performed by Bontempo et al. [10], who explain their observed values by the presence of volcanic sulfur. As we are unfamiliar with Italian geology and the information about where and how these samples were taken, it is difficult for us to make other conclusions. Based on this study, we can suggest that a possible source of S could be oxidized (biogenic) sulfide in sedimentary rocks since volcanic rocks usually possess *δ*^34^S values above the threshold value [11].

Further, the *δ*^34^S were found to vary in the buckwheat sub-samples. However, we believe this is not only due to the low S-concentration, although the uncertainty of stable isotope measurements at lower S-concentrations is generally higher, and some deviation in *δ*^34^S values can occur. It should be noted that the authors of the original article have broad experience in metrology and evaluating stable isotope measurement. This experience comes from years of participating in interlaboratory comparisons Food analysis using Isotopic Techniques–Proficiency Testing Scheme (FIT-PTS) in CCQM (Comité Consultatif pour la Quantite de Matière) and developing food matrix reference materials, including nitrogen and sulfur stable isotope-ratio measurements. Based on this experience, we made assumptions about the determined *δ*^34^S values [12]. Furthermore, as the authors of the comment are aware, using sulfur (S) stable isotopes to study S metabolism in plants is still limited. Although it is generally accepted that less S stable isotope discrimination occurs during sulfate (SO_4_^2−^) uptake, S metabolism and allocation are expected to produce separations of stable S isotopes among the different plant S pools and different tissues. For example, the latest study on rice by Cavallaro et al. [13] shows that S_org_ pools in the rice were ^34^S-depleted compared to the S-SO_4_^2−^ source. Our results similarly indicate that S_org_ (in semolina) is ^34^S-depleted compared to hulls containing more inorganic S. However, the exact mechanisms need further investigation. Again, it should be noted that: “*We are aware that the number of samples is limited; however, we believe that the information provided by the paper is essential for establishing an appropriate database of authentic buckwheat samples that, to our knowledge, does not currently exist*.”

Finally, as noted by Dr’s Horacek and Cannavan, the *δ*^34^S data in Table 2 raws 2 and 11 should be corrected as follows 7.2 and 5.5‰, respectively. Consequently, the *δ*^34^S range in Table 2 on lines 9 and 22 should be corrected as follows: 3.6–7.2‰ and 3.6–5.5‰, respectively.

## Data Availability

The data presented in this study are available from the corresponding author upon request.

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
