# Peer review of "Reply to Horacek, M.; Cannavan, A. Comment on “Sinkovič et al. Isotope Fingerprints of Common and Tartary Buckwheat Grains and Milling Fractions: A Preliminary Study. Foods 2022, 11, 1414”"

_foods, 2022, doi:10.3390/foods11172628_

Round 1
Reviewer 1 Report
The authors in this ''reply text'' have documented in details that the isotope fingerprint of common and tartary buckwheat grains is affected by the pedoclimatic conditions in its geographical origin and the processing techniques applied. That is why the different milling fractions have adverse isotopic data. I am curious about the need of such reply? Is it mandatory?
From a scientific point of view the text I received is ok. Minor spell check is required. I have attached the pdf with a minor correction.
Based on the lack of problems, I suggest publication of this replying text after a minor revision.

Author Response
"Please see the attachment."

Reviewer 2 Report
No.
Author Response
"Please see the attachment."
